# Long COVID in the United States

**David G. Blanchflower**[1,2,3☯], **Alex Bryson**[4,5☯] *

**1** Department of Economics, Dartmouth College, Hanover, NH, United States of America, **2** Adam Smith Business School, University of Glasgow, Glasgow, United Kingdom, **3** NBER, Cambridge, MA, United States of America, **4** UCL Social Research Institute, University College London, London, United Kingdom, **5** NIESR and IZA, London, United Kingdom

☯ These authors contributed equally to this work.
* a.bryson@ucl.ac.uk

**Data Availability Statement:** The data can be acquired via the US Census Bureau. The data are publicly available and can be downloaded here: https://www.census.gov/programs-surveys/household-pulse-survey/datasets.html. The accompanying technical documentation can be

## Abstract

Although yet to be clearly identified as a clinical condition, there is immense concern at the health and wellbeing consequences of long COVID. Using data collected from nearly half a million Americans in the period June 2022-December 2022 in the US Census Bureau's Household Pulse Survey (HPS), we find 14 percent reported suffering long COVID at some point, half of whom reported it at the time of the survey. Its incidence varies markedly across the United States–from 11 percent in Hawaii to 18 percent in West Virginia–and is higher for women than men, among Whites compared with Blacks and Asians, and declines with rising education and income. It is at its highest in midlife in the same way as negative affect. Ever having had long COVID is strongly associated with negative affect (anxiety, depression, worry and a lack of interest in things), with the correlation being strongest among those who currently report long COVID, especially if they report severe symptoms. In contrast, those who report having had short COVID report *higher* wellbeing than those who report never having had COVID. Long COVID is also strongly associated with physical mobility problems, and with problems dressing and bathing. It is also associated with mental problems as indicated by recall and understanding difficulties. Again, the associations are strongest among those who currently report long COVID, while those who said they had had short COVID have fewer physical and mental problems than those who report never having had COVID. Vaccination is associated with lower negative affect, including among those who reported having had long COVID.

## 1. Introduction

The SARS-CoV-2 (or COVID) pandemic has resulted in an estimated 6.8 million deaths around the world since the beginning of the outbreak in December 2019 and severely impacted the lives of many others who suffered temporary illness [1]. Fig 1 plots the weekly estimates of COVID cases and deaths in the United States reported by the Centers for Disease Control and Prevention (CDC) [2]. These show three major spikes in cases, peaking at 5.6 million at the start of 2022. The number of deaths has four major peaks with the highest, 23,387, again at the start of 2021. The latest estimate for the start of 2023, was 415,000 cases and 3,900 deaths. However, it has become increasingly apparent that a significant proportion of the population

found here: https://www.census.gov/programs-surveys/household-pulse-survey/technical-documentation.html.

**Funding:** The authors received no specific funding for this work.

**Competing interests:** The authors have declared that no competing interests exist.

continue to report COVID symptoms long after initial infection. This condition, which has come to be known as 'long COVID', has yet to be clearly identified as a clinical condition, but is defined by the World Health Organization (WHO) as the continuation or development of new symptoms three months after the initial infection, with these symptoms lasting for at least 2 months with no other explanation [3].

The recency of long COVID, together with difficulties in precisely defining it [4] mean uncertainty persists regarding its incidence and consequences. Notwithstanding these issues, a consensus is emerging regarding its incidence, factors associated with it, and its health consequences. We review this literature below in Section Two. We contribute to this body of work by exploiting new survey data on the incidence and consequences of long COVID in the US Census Bureau's Household Pulse Survey (HPS) [5]. In doing so we build on an earlier paper [6] that examined the incidence of COVID and changes in mental health over the period April 2020-April 2022. That paper used data from HPS sweeps #1-#44. This paper extends that work from June through December 2022 with new data on long COVID from sweeps #46-#53 that was not available in earlier sweeps.

We find 14 percent of respondents reported suffering long COVID at some point, half of whom reported it at the time of the survey. Its incidence varies markedly across the United

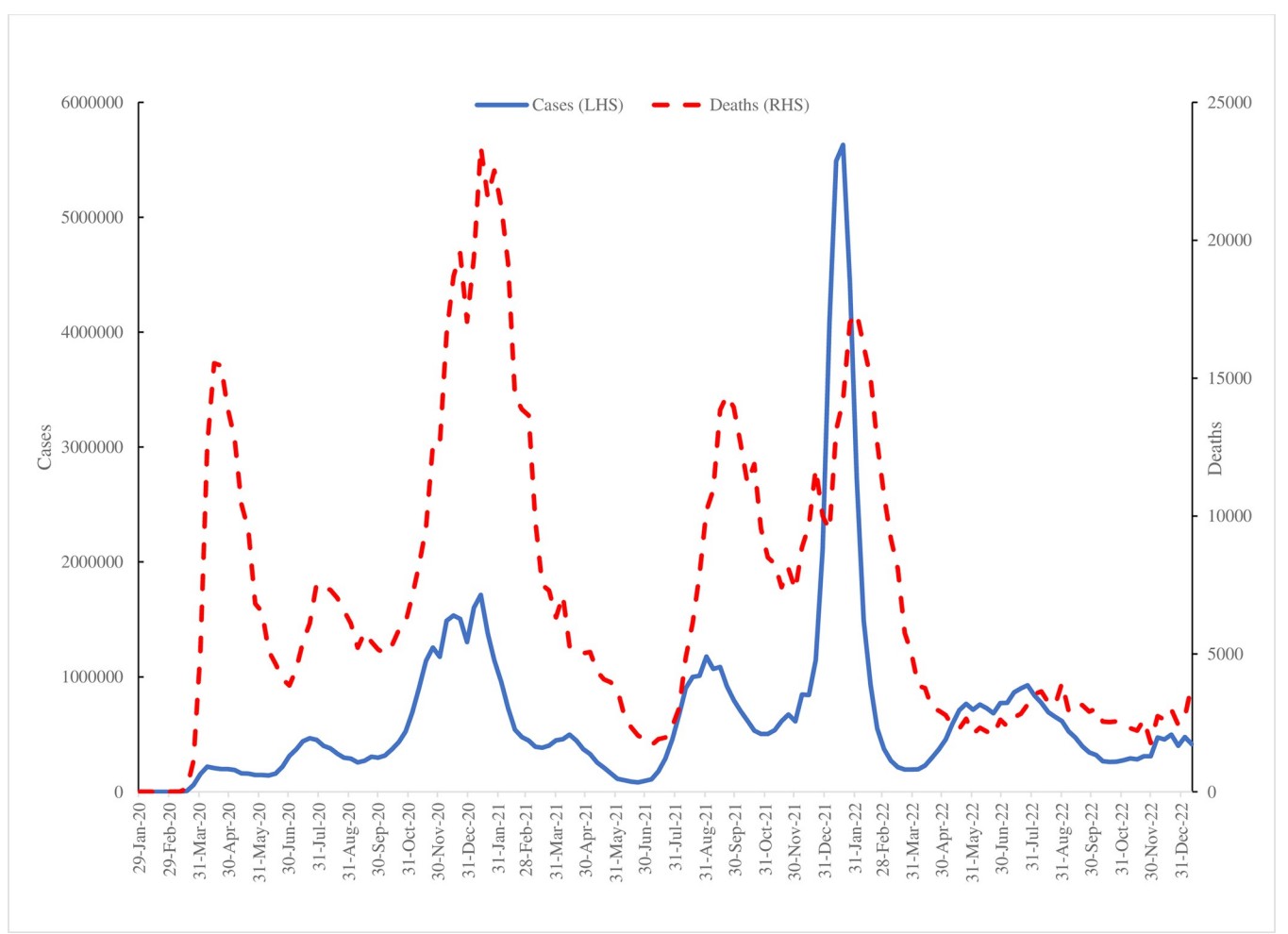

**Fig 1. Number of COVID cases and death per week in the United States.**

States–from 11 percent in Hawaii to 18 percent in West Virginia–and is higher for women than men, among Whites compared with Blacks and Asians, and declines with rising education and income. It is at its highest in midlife in the same way as negative affect. Ever having had long COVID is strongly associated with negative affect (anxiety, depression, worry and a lack of interest in things), with the correlation being strongest among those who currently report long COVID, especially if they report severe symptoms. In contrast, those who report having had short COVID report *higher* wellbeing than those who report never having had COVID. Long COVID is also strongly associated with physical mobility problems, and with problems dressing and bathing. It is also associated with mental problems as indicated by recall and understanding difficulties. Again, the associations are strongest among those who currently report long COVID, while those who said they had had short COVID have fewer physical and mental problems than those who report never having had COVID. Vaccination is associated with lower negative affect, including among those who reported having had long COVID.

## 2. Previous literature on the incidence and health effects of long COVID

In their systematic review article Davis et al. [7] define long COVID as "a multisystemic condition comprising often severe symptoms that follow a sever acute respiratory syndrome coronavirus 2 (SARS-CoV-2) infection" (p. 134). They say that more than 200 symptoms have been identified, with impacts on multiple organ systems. They estimate that "at least 10%" of those with severe COVID infections go on to develop long COVID. This is roughly 65 million individuals worldwide. They say many long COVID sufferers experience "dozens of symptoms across multiple organ systems" (p. 134). Among them are cognitive and physical impairments we focus on in this study.

In their systematic review of 194 studies–most of them in Europe—O'Mahoney et al. [8] showed that at an average follow-up time of 126 days, 45% of COVID survivors, regardless of hospitalization status, go on to experience at least one unresolved symptom. In addition, the prevalence of ongoing symptoms appears to be higher in post-hospitalized cohorts compared to non-hospitalized populations. Taquet et al. [9] conducted a retrospective cohort study based on linked electronic health records (EHRs) data from 81 million patients including 273,618 COVID survivors and found 57% of patients having at least one long-COVID feature recorded within the first 180 days after infection and 37% having them in the 90 to 180 days after diagnosis.

Correlates of long COVID emerging from this literature include variance by age, and a potential role for vaccination which, at least in some studies, reduced the probability of long COVID [7, p. 140]. Taquet et al. [9] found a higher incidence of long COVID among the elderly, in more severely affected patients, and in women, patterns also found by Subramanian et al. [10] among non-hospitalized adults. Qasmieh et al. [11] also found long COVID was higher among females, but also among Blacks and the unemployed. Price [12], who used the HPS as we do, found long-term COVID symptoms were much more prevalent among women, adults under 65, Hispanics and Latinos, and non–college graduates than among other demographic groups. Using a non-probability internet survey and a slightly different definition of long COVID (symptomatic two months after initial infection) Perlis et al. [13] also found a higher incidence among women. They also pointed to a lower incidence among the more highly educated and among those who had been vaccinated. Symptoms included cognitive and respiratory problems, anxiety and sleep disruption.

For the UK the Office for National Statistics calculated the incidence of long COVID from their Coronavirus (COVID) Infection Survey [14]. They estimated that 2.1 million people

living in private households in the UK (3.3% of the population) were experiencing self-reported long COVID (which they defined as symptoms continuing for more than four weeks after the first confirmed or suspected COVID infection that were not explained by something else) as of 4 December 2022. Long COVID symptoms adversely impacted the day-to-day activities of 1.6 million. The prevalence was greatest in midlife ages 35–69.

Using the same ONS survey Ayoubkhani et al. [15] found the odds of long COVID symptoms decreased after COVID vaccination by around 13 percent, while a second dose reduced the odds by around 9% thus sustaining protection against long COVID. In a related study with the same survey Ayoubkhani et al. [16] find long COVID symptoms were significantly lower among those receiving 2 doses of the vaccine after 12 weeks relative to being unvaccinated when infected.

Also, for the UK, Bagues and Dimitrova [17] report psychological gains from COVID vaccination. Although their paper is not focused on long COVID, it is notable in using exogenous variance in the timing of vaccination roll-out to make causal inferences about the positive impact of COVID vaccination in increasing psychological wellbeing as captured by a reduction in the General Health Questionnaire short form (GHQ-12) scale capturing mental distress.

Notwithstanding differences in the precise definition of long COVID, survey design and populations, taken together, these previous studies indicate that long COVID is a common condition among those infected with the virus, although its incidence varies across demographic groups. It often comes with multiple symptoms which can be severe and affect both cognitive and physical function.

In the next section of the paper we describe our data and methods. In the subsequent section we present the incidence of long COVID in the population of the United States, and its incidence across sub-populations before estimating the probability of long COVID, thus identifying the independent correlations with a number of individual traits, as well as location and vaccination status. In the final part of Section Four we establish the association between long COVID and cognitive and physical impairment as well as ill-being. Section Five concludes.

## 3. Data and methods

Our data are the United States Census Bureau's Household Pulse Survey (HPS) for the period January through December 2022 [5]. The HPS was designed to obtain data on how people's lives had been impacted by COVID in a quick and efficient manner. It does so with a short online survey. These sweeps of the data include new data on long COVID from sweeps June 2022 onwards (sweeps #46-#53) that was not available in earlier sweeps [18]. The data we examine here on long COVID is from sweeps #46 (Jun 1-Jun 13, 2022);, #47 (June 29-July 11, 2022);; #48 (July 27-Aug 8, 2022);; #49 (Sept 14-Sept 26, 2022);; #50 (Oct 5-Oct 17, 2022); #51 (Nov 2-Nov 14, 2022);, #52 (Dec 9-Dec 19, 2022); and #53 (Jan 4–16, 2023). They form part of an extended battery of 14 questions on COVID infection (see S1 Appendix). The sample size is 461,550.

The data are publicly available and can be downloaded here: https://www.census.gov/programs-surveys/household-pulse-survey/datasets.html The accompanying technical documentation can be found here: https://www.census.gov/programs-surveys/household-pulse-survey/technical-documentation.html

In Section Four we report simple descriptive statistics establishing the incidence of long COVID in the United States population. In doing so we weight the descriptive statistics with the person weight variable (PWEIGHT) provided by the US Census to account for non-response biases. We then run regression analyses to establish the correlates of long COVID, long COVID experienced now, and long COVID with substantial symptoms, using

unweighted data. All three are (0,1) outcomes and so are estimated with probit regressions. We then turn to the association between long COVID now and in the past on four negative affect measures with ordinal responses, so we estimate the long COVID effect using ordinal logits. A composite negative affect metric adding up the scores on the four separate negative affect measures is estimated with Ordinary Least Squares (OLS) estimation.

## 4. Results

### 4.1: Incidence of long COVID and long COVID symptoms in the US Census HPS

Whereas almost half (46.7%) of respondents say they have had COVID at some point ('yes' to Q2 in S1 Appendix) only 14.4% said they had ever had COVID symptoms lasting 3 months or longer (Q3). This suggests that around three-in-ten of those who get COVID go on to develop long COVID. There were 6,844 unclassified, unweighted cases due to missing values and 59,505 (unweighted) cases with long COVID of whom 29,839 had symptoms now.

The survey prompted respondents as to what long term symptoms they may have, prompting them with the following description:

Did you have any symptoms lasting 3 months or longer that you did not have prior to having coronavirus or COVID? Long term symptoms may include—tiredness or fatigue, difficulty thinking, concentrating, forgetfulness, or memory problems (sometimes referred to as "brain fog", difficulty breathing or shortness of breath, joint or muscle pain, fast-beating or pounding heart (also known as heart palpitations), chest pain, dizziness on standing, menstrual changes, changes to taste/smell, or inability to exercise? (Q1)

Those who had had COVID were asked about the severity of their symptoms. 13.3% said they had suffered "severe symptoms" (S1 Appendix, Question 5). Table 1 shows how severe the symptoms of COVID were for those who had had short COVID–that is, COVID symptoms lasting less than 3 months—and those who reported long COVID now and in the past. Of those with short COVID only 7% had severe symptoms compared with 24% for those who had had long COVID in the past and 31% who currently had long COVID. Even so, this means two-thirds of those with long COVID now were not suffering severe symptoms.

In sweeps #49-#53 which cover the period September 14 2022 to January 16 2023—a further question was asked of those with long COVID–Q6. *Do these long-term symptoms reduce your ability to carry out day-to-day activities compared with the time before you had COVID*? The sample size across these five sweeps is 288,951. 6.9% of weighted respondents currently reporting long COVID also reported such symptoms (*Longnow)*. We decided to classify as *Longlot* those who said they had long COVID now and their symptoms reduced their ability to carry out day-to-day activities 'a lot'. Overall, 1.6% of the sample from weeks 49–53 or a quarter of those who say they have COVID, now have major debilitating issues.

**Table 1. The distribution of symptoms (n = unweighted; figures are weighted column percentages).**

|  | No COVID | Short COVID | Long COVID not now | Long COVID now | All | N |
|---|---|---|---|---|---|---|
| No COVID |  |  |  |  | 53 | 244,414 |
| No symptoms |  | 8 | 3 | 2 | 3 | 10,382 |
| Mild symptoms |  | 47 | 26 | 21 | 19 | 85,826 |
| Moderate symptoms |  | 38 | 48 | 46 | 19 | 88,652 |
| Severe symptoms |  | 7 | 24 | 31 | 6 | 24,189 |
| All row % | 53 | 33 | 8 | 7 |  |  |
| N | 244,414 | 149,732 | 29,510 | 29,807 | 453,463 | 453,463 |

Our finding that 6.9% had long COVID with current symptoms is consistent with results from a recent paper by McKaylee et al. [19] who estimated the prevalence of long COVID from June 30-July 2, 2022, in a random sample of 3,042 adults from the US population. They found 7.3% reporting long COVID.

Table 2 presents means for ever having had long COVID (*LongCOVID)* and currently having long COVID with current symptoms (*Longnow)* in parentheses for the whole of the United States and for sub-populations. The patterns are similar for both. The incidence of long COVID is greatest in midlife, it is higher for women (as found by Chen et al. [20]) and declines with education and income. Long COVID is also more prevalent among whites than blacks or Asians. There is substantial regional variance across States: Hawaii has lowest incidence and West Virginia highest.

The CDC has reported the incidence of long COVID using these data for each of the seven surveys we examine. They found the incidence higher among women, the prime age, Hispanics and high school dropouts and highest in West Virginia, Wyoming, Mississippi and Kentucky and lowest in Vermont and Rhode Island. https://www.cdc.gov/nchs/COVID19/pulse/long-COVID.htm

## 4.2: The probability of having long COVID

To establish independent associations between long COVID and individuals' characteristics we run a set of probit estimates in Table 3. The covariates entering the estimates are: age (15 bands), race (5 categories), male, education (7 categories), whether the respondent was working, state of residence and week of interview.

In column 1 of Table 3 we estimate a probit equation for the (0,1) outcome that an individual has ever had long COVID (*LongCOVID*). The second column presents probit estimates for the (0,1) outcome of having long COVID *now* with symptoms (*Longnow*). The final column reports estimates for the (0,1) outcome of long COVID now with symptoms that reduce one's ability to carry out day-to-day activities 'a lot' (*Longlot*). The sample size is smaller for the final column because the question on symptoms affecting one's ability to carry out day-to-day activities is only asked between September 14 2022 and January 16 2023.

Consistent with the existing studies, women are more likely to suffer long COVID. They are also more likely than men to report currently having long COVID with symptoms (column 2), and to have symptoms that affect them a lot (column 3). Whites are more likely to suffer long COVID than non-white ethnic groups. They are also more likely to have long COVID with symptoms and symptoms that affect their day-to-day activities 'a lot'.

The incidence of long COVID is highest in Alabama, Mississippi and West Virginia–which are also among the states with the lowest subjective wellbeing rankings in the United States [21] whilst Hawaii—the highest ranked state on subjective wellbeing [21]–has the lowest incidence of long COVID.

The probability of long COVID falls among graduates, as does the probability of long COVID with symptoms, and with severe symptoms.

In their earlier study, Blanchflower and Bryson [6] found workers were more likely to have had COVID than non-workers. The simple descriptive means in Table 2 indicate that workers were more likely than non-workers to have had long COVID and to have long COVID with symptoms at the time of interview. However, there is no statistically significant association between working and having had long COVID in Table 3 (the coefficient in column 1 is 0.002 with a t-statistic of 0.35). Furthermore, working was negatively associated with currently having long COVID with symptoms (column 2) and currently having long COVID with symptoms that affected their day-to-day activity a lot (column 3). This may be because individuals

**Table 2. Weighted mean proportion with long COVID and long COVID now with symptoms.**

| | | | |
|---|---|---|---|
| USA | 14.4 (6.9) | Michigan | 13.8 (6.7) |
| Men | 11.1 (5.0) | Minnesota | 13.3 (6.5) |
| Women | 17.6 (8.7) | Mississippi | 17.6 (8.3) |
| Age <30 | 15.8 (6.4) | Missouri | 14.7 (7.4) |
| Age 30–39 | 16.4 (7.4) | Montana | 15.8 (8.7) |
| Age 40–49 | 17.6 (8.3) | Nebraska | 14.3 (6.6) |
| Age 50–59 | 15.9 (8.1) | Nevada | 15.3 (6.9) |
| Age 60–69 | 11.2 (6.1) | New Hampshire | 12.6 (6.5) |
| Age 70–79 | 7.9 (4.5) | New Jersey | 13.1 (6.3) |
| Age 80+ | 6.4 (3.5) | New Mexico | 15.8 (7.5) |
| Hispanic | 18.6 (7.6) | New York | 14.5 (6.8) |
| Asian | 9.8 (4.0) | North Carolina | 14.6 (6.8) |
| Black | 12.8 (5.7) | North Dakota | 15.9 (7.7) |
| Non-Hispanic white | 13.8 (7.0) | Ohio | 13.7 (7.1) |
| Non-Hispanic other | 18.0 (9.0) | Oklahoma | 17.1 (8.1) |
| Less than high school | 16.5 (9.6) | Oregon | 13.1 (6.5) |
| High school diploma | 14.3 (6.4) | Pennsylvania | 12.8 (5.5) |
| Some college | 16.9 (8.4) | Rhode Island | 13.8 (7.0) |
| Bachelor's degree | 12.5 (6.1) | South Carolina | 15.0 (7.8) |
| Graduate degree | 10.4 (5.1) | South Dakota | 15.8 (7.7) |
| Working | 15.2 (7.1) | Tennessee | 16.8 (8.3) |
| Not working | 13.5 (6.6) | Texas | 15.3 (7.1) |
| Alabama | 17.5 (8.8) | Utah | 16.4 (8.0) |
| Alaska | 16.8 (7.7) | Vermont | 10.8 (5.9) |
| Arizona | 16.6 (7.9) | Virginia | 12.1 (5.7) |
| Arkansas | 17.4 (8.6) | Washington | 12.5 (6.7) |
| California | 14.0 (6.7) | West Virginia | 18.2 (9.3) |
| Colorado | 15.8 (7.3) | Wisconsin | 13.8 (6.9) |
| Connecticut | 13.2 (6.7) | Wyoming | 18.0 (8.3) |
| Delaware | 11.5 (6.2) | Vaccinated | 14.2 (6.9) |
| District of Columbia | 9.8 (5.1) | Less than $25,000 | 16.5 (8.0) |
| Florida | 13.7 (6.2) | $25,000 - $34,999 | 16.9 (8.6) |
| Georgia | 14.6 (6.9) | $35,000 - $49,999 | 16.2 (8.2) |
| Hawaii | 11.0 (4.0) | $50,000 - $74,999 | 15.2 (7.5) |
| Idaho | 16.2 (8.3) | $75,000 - $99,999 | 14.5 (7.2) |
| Illinois | 14.2 (6.5) | $100,000 - $149,999 | 13.4 (6.6) |
| Indiana | 15.8 (7.6) | $150,000 - $199,999 | 11.8 (5.9) |
| Iowa | 14.6 (7.7) | $200,000 and above | 9.3 (4.3) |
| Kansas | 14.8 (7.0) | | |
| Kentucky | 15.9 (8.4) | Figures in parentheses % long COVID now. | |
| Louisiana | 16.1 (7.3) | | |
| Maine | 12.0 (6.4) | | |
| Maryland | 12.2 (5.6) | | |
| Massachusetts | 12.7 (6.1) | | |
| Michigan | 13.8 (6.7) | | |

**Table 3. Probit estimates of long COVID, long COVID with symptoms now and ever had COVID 2022.**

|  | Long COVID | Longnow | Longlot |
|---|---|---|---|
| 20–24 | .2133 (5.54) | .2489 (4.72) | .3988 (3.19) |
| 25–29 | .2190 (5.88) | .2557 (4.99) | .3496 (2.84) |
| 30–34 | .2331 (6.31) | .2887 (5.68) | .4032 (3.31) |
| 35–39 | .2534 (6.89) | .3447 (6.81) | .4858 (4.00) |
| 40–44 | .2853 (7.76) | .3871 (7.66) | .5428 (4.48) |
| 45–49 | .2886 (7.84) | .4269 (8.44) | .6134 (5.07) |
| 50–54 | .2565 (6.97) | .4039 (7.99) | .5815 (4.80) |
| 55–59 | .1904 (5.17) | .3591 (7.10) | .5808 (4.80) |
| 60–64 | .0671 (1.82) | .2735 (5.41) | .4725 (3.91) |
| 65–69 | -.0766 (2.08) | .1350 (2.66) | .2559 (2.11) |
| 70–74 | -.1707 (4.57) | .0538 (1.05) | .1493 (1.22) |
| 75–79 | -.2412 (6.27) | -.0074 (0.14) | .1347 (1.08) |
| 80–84 | -.2469 (5.91) | -.0010 (0.02) | .2040 (1.57) |
| 85–89 | -.2605 (5.30) | -.0195 (0.31) | .3022 (2.16) |
| Male | -.2883 (56.70) | -.2833 (15.14) | -.1744 (12.96) |
| Some high school | -.0277 (0.80) | -.1429 (3.37) | -.2329 (3.26) |
| High school graduate | -.0351 (1.20) | -.0875 (2.50) | -.2987 (5.16) |
| Some college | .0290 (1.01) | .0145 (0.42) | -.2119 (3.74) |
| Associate degree | .0290 (0.99) | .0136 (0.39) | -.2220 (3.83) |
| Bachelor degree | -.1692 (5.86) | -.1480 (4.28) | -.4124 (7.24) |
| Graduate degree | -.2462 (8.49) | -.2108 (6.07) | -.4390 (7.63) |
| Black | -.2305 (18.34) | -.1728 (11.02) | -.0875 (2.65) |
| Asian | -.3396 (22.09) | -.2961 (14.89) | -.2541 (5.43) |
| Other | -.0007 (0.06) | .0918 (5.69) | .1986 (6.14) |
| White non-Hispanic | -.1665 (17.75) | -.0652 (5.60) | -.0542 (2.17) |
| Work | .0020 (0.35) | -.0199 (2.84) | -.2848 (19.76) |
| Alaska | -.1358 (4.81) | -.1229 (3.62) | -.2285 (3.17) |
| Arizona | -.1627 (6.62) | -.1638 (5.52) | -.1929 (3.11) |
| Arkansas | -.0309 (1.13) | -.0453 (1.38) | .0152 (0.24) |
| California | -.2602 (11.91) | -.2157 (8.22) | -.2009 (3.71) |
| Colorado | -.1497 (6.04) | -.1411 (4.72) | -.1823 (2.87) |
| Connecticut | -.2067 (7.51) | -.1433 (4.36) | -.1299 (1.88) |
| Delaware | -.2364 (7.36) | -.1866 (4.84) | -.1352 (1.71) |
| DC | -.4132 (13.34) | -.3234 (8.59) | -.2843 (3.45) |
| Florida | -.2101 (8.72) | -.1997 (6.86) | -.1502 (2.51) |
| Georgia | -.1545 (6.18) | -.1407 (4.66) | -.2096 (3.24) |
| Hawaii | -.3372 (10.40) | -.3504 (8.66) | -.3582 (4.07) |
| Idaho | -.0872 (3.35) | -.0748 (2.40) | -.0773 (1.24) |
| Illinois | -.1821 (7.11) | -.1715 (5.53) | -.1839 (2.80) |
| Indiana | -.0962 (3.67) | -.1138 (3.59) | -.0907 (1.39) |
| Iowa | -.1504 (5.64) | -.1214 (3.79) | -.1294 (1.95) |
| Kansas | -.1251 (4.81) | -.1351 (4.30) | -.1645 (2.47) |
| Kentucky | -.0972 (3.60) | -.0797 (2.46) | -.2369 (3.35) |
| Louisiana | -.0708 (2.53) | -.1118 (3.27) | -.0350 (0.52) |
| Maine | -.2503 (7.95) | -.1986 (5.26) | -.2161 (2.68) |
| Maryland | -.2683 (10.25) | -.2537 (7.95) | -.2746 (3.97) |
| Massachusetts | -.2397 (9.54) | -.2122 (6.99) | -.1798 (2.79) |

*(Continued)*

**Table 3.** (Continued)

| | Long COVID | Longnow | Longlot |
|---|---|---|---|
| Michigan | -.1538 (6.30) | -.1401 (4.77) | -.0911 (1.53) |
| Minnesota | -.2179 (8.50) | -.1799 (5.84) | -.1657 (2.56) |
| Mississippi | .0051 (0.18) | -.0118 (0.34) | -.0971 (1.36) |
| Missouri | -.1454 (5.54) | -.1158 (3.68) | -.0827 (1.29) |
| Montana | -.0546 (1.90) | -.0664 (1.92) | .0177 (0.27) |
| Nebraska | -.1372 (5.13) | -.1588 (4.88) | -.2596 (3.68) |
| Nevada | -.1435 (5.17) | -.1341 (4.01) | -.2307 (3.16) |
| New Hampshire | -.2063 (7.30) | -.1670 (4.93) | -.1448 (2.02) |
| New Jersey | -.1733 (6.45) | -.1438 (4.44) | -.1626 (2.36) |
| New Mexico | -.1668 (6.26) | -.1310 (4.10) | -.1694 (2.57) |
| New York | -.1521 (5.70) | -.1443 (4.47) | -.1726 (2.50) |
| North Carolina | -.1723 (6.44) | -.1327 (4.14) | -.0952 (1.44) |
| North Dakota | -.0823 (2.71) | -.1144 (3.09) | -.1780 (2.27) |
| Ohio | -.1619 (6.01) | -.1374 (4.24) | -.1246 (1.85) |
| Oklahoma | -.0426 (1.63) | -.0584 (1.86) | -.0884 (1.38) |
| Oregon | -.2712 (10.90) | -.2282 (7.63) | -.1550 (2.53) |
| Pennsylvania | -.2189 (8.73) | -.2097 (6.90) | -.1845 (2.89) |
| Rhode Island | -.1624 (4.98) | -.1346 (3.43) | -.1726 (2.08) |
| South Carolina | -.1059 (3.92) | -.1013 (3.12) | -.1131 (1.68) |
| South Dakota | -.0945 (3.23) | -.0945 (2.68) | -.1020 (1.41) |
| Tennessee | -.0434 (1.68) | -.0308 (1.00) | -.0776 (1.21) |
| Texas | -.1824 (8.13) | -.1643 (6.08) | -.1361 (2.47) |
| Utah | -.1028 (4.22) | -.0950 (3.24) | -.0813 (1.35) |
| Vermont | -.3122 (9.63) | -.2118 (5.53) | -.2112 (2.51) |
| Virginia | -.2902 (11.57) | -.2484 (8.19) | -.2105 (3.32) |
| Washington | -.2943 (12.62) | -.2113 (7.58) | -.1239 (2.19) |
| West Virginia | .0232 (0.80) | .0280 (0.81) | .0097 (0.14) |
| Wisconsin | -.1885 (7.08) | -.1737 (5.39) | -.2187 (3.16) |
| Wyoming | -.0352 (1.21) | -.0463 (1.33) | -.0968 (1.34) |
| Constant | -.7674 | -1.3302 | -1.8703 |
| Pseudo $R^2$ | .0406 | .0309 | .0419 |
| N | 448,227 | 448,119 | 285,954 |

Notes: Reference categories are: Alabama; less than high school; age<20 and white Hispanic. T-statistics in parentheses. Controls also include week of interview. Pseudo $R^2$ is a measure of model fit. *Longlot* only available in weeks #49-#53

with long COVID with symptoms felt obliged to stay away from work for fear of spreading the virus, or because they were physically unable to perform work tasks and thus forced onto temporary or long-term sickness absence.

The age structure of long COVID tracks that of the unhappiness literature peaking in mid-life between the ages of 45 and 49 [22]. This contrasts with the findings in our earlier paper regarding the incidence of COVID [6] which peaked between the ages of 20–24. Long COVID incidence ever and at the time of interview peaked in mid-life.

Overall, in the sample 83.2% of respondents had received a vaccine. Of those who had not had Covid 84.2% had received a vaccine, compared with 82.0% who had had short covid and 82.1% of those who reported they at some time suffered from long covid.

## 4.3: The impact of long COVID on wellbeing and physical and mental health

In this section we examine the impact of long COVID and symptoms on four negative affect measures–anxiety, worry, being down and depressed, and showing a lack of interest or pleasure in doing things, as indicated by responses to the following questions:

Over the last 2 weeks, how often have you been bothered by:

a. *feeling nervous, anxious, or on edge*?

b. *not being able to stop or control worrying*?

c. *by feeling down, depressed, or hopeless*?

d. *bothered by having little interest or pleasure in doing things*?

Answers were coded as Not at all = 1, Several days = 2, More than half the days = 3 and Nearly every day = 4.

These are Questions 7 to 10 in S1 Appendix. The S1 Appendix also provides the weighted percentages for each response category.

The four variables follow similar time series paths. Following Blanchflower and Bryson [6] in Fig 2 we plot a composite score which sums these four variables and runs from four to

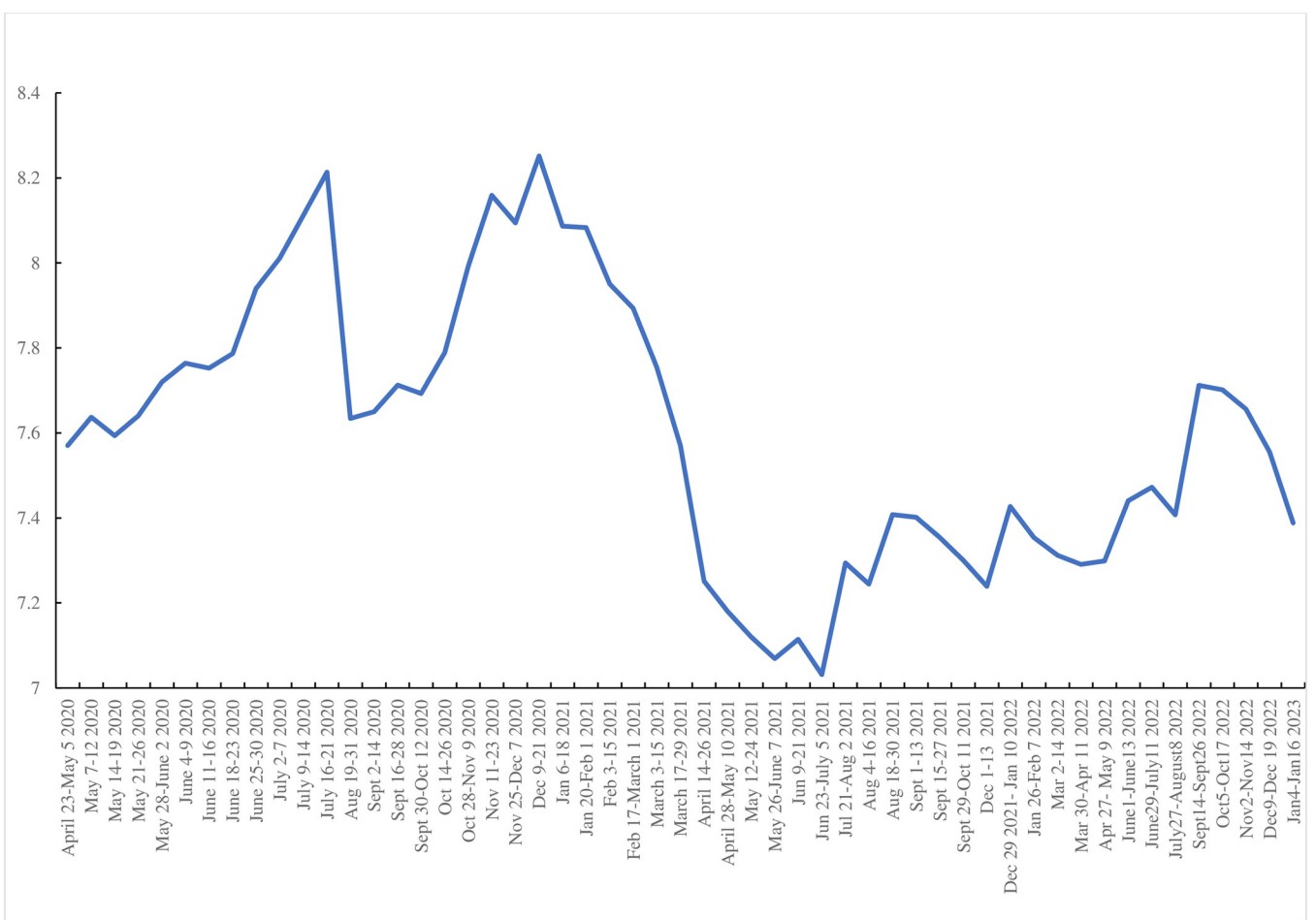

**Fig 2. Composite negative affect index, April 2020-January 2023.**

**Table 4. Raw mean scores for negative affect variables and mobility and cognizance variables by having long or short COVID–n = 454,706.**

|  | No covid | Short COVID | Had long COVID | Long COVID now |
|---|---|---|---|---|
| Panel A: |  |  |  |  |
| Anxious | 1.94 | 1.87 | 2.28 | 2.53 |
| Down & depressed | 1.76 | 1.66 | 1.99 | 2.25 |
| Worry | 1.86 | 1.78 | 2.14 | 2.42 |
| Interest | 1.79 | 1.68 | 2.02 | 2.30 |
| Combined | 7.43 | 7.06 | 8.48 | 9.61 |
| Panel B: |  |  |  |  |
| Mobility | 1.37 | 1.23 | 1.36 | 1.60 |
| Self-care | 1.12 | 1.07 | 1.12 | 1.21 |
| Understand | 1.11 | 1.07 | 1.12 | 1.20 |
| Remember | 1.49 | 1.44 | 1.68 | 1.96 |

sixteen where 4 would be scored if the respondent answered "Not at all" to all four items whereas 16 would be scored by responding "Nearly every day" on all four. It has a weighted mean of 7.54. It shows a sharp pick up in weeks #49 (September 14–26, 2022) and #50 (October 5–17, 2022) and then a subsequent decline. Referring back to Fig 1 this seems to be associated with the peak in deaths at the end of August 2022.

The first four rows of Table 4 report mean scores for the four separate affect variables, together with the composite index in row 5 (labelled "Combined"), for those with who have never had COVID (column 1), those who have had short COVID at some point (column 2), those who had long COVID in the past, and those who were experiencing long COVID at the time of the survey.

The table below shows their incidence in the data:

|  | Weighted % |
|---|---|
| Never had COVID | 52.4 |
| Short COVID: COVID but not long COVID | 33.2 |
| Had long COVID no symptoms now | 7.5 |
| Had long COVID with current symptoms | 6.9 |

It is apparent that there is little difference in the negative affect experienced by those who had never suffered from COVID and those who had only experienced short COVID. However, those who had ever experienced long COVID experienced greater negative affect, particularly if they were currently experiencing long COVID.

In Panel B of Table 4 we report mean scores for mobility difficulties ('difficulty walking or climbing stairs' (Q11), difficulties with self-care such as washing or dressing (Q12), understanding or being understood (Q13) and difficulties remembering or concentrating (Q14). In each case, these difficulties are coded from 1 ("No difficulty") to 4 ("Cannot do at all"). In all cases the same pattern is apparent: physical and mental difficulties are greater for those who had experienced long COVID, particularly those currently experiencing it.

In Table 5 we run ordered logit regressions to estimate the association between COVID status and the four negative affect ordered outcomes, controlling for other factors. The last column reports OLS estimates from the (4,16) continuous outcome based on the summed total of the four negative affect variables. The distribution of this variable is as follows weighted percent—4 = 31.0%; 5 = 8.5; 6 = 9.4; 7 = 8.3; 8 = 12.4; 9 = 5.4; 10 = 4.6; 11 = 3.3; 12 = 4.1; 13 = 2.6; 14 = 2.5; 15 = 2.0 and 16 = 5.9.

Table 5. Regression estimates for 5 negative affect measures.

|  | Anxious | Down | Worry | Interest | Combination (OLS) |
|---|---|---|---|---|---|
| Short COVID | -.2677 (39.47) | -.3095 (42.94) | -.2587 (36.82) | -.3051 (42.54) | -.5609 (48.09) |
| Long COVID not now | .3168 (25.91) | .2557 (20.23) | .3236 (26.10) | .2884 (22.98) | .5691 (26.13) |
| Long COVID now | .7943 (66.29) | .7417 (60.92) | .7525 (62.43) | .7924 (65.52) | 1.6261 (76.76) |
| Male | -.4073 (66.51) | -.1775 (27.56) | -.4539 (71.50) | -.0815 (12.74) | -.4515 (42.89) |
| In paid work | -.2119 (28.62) | -.3268 (42.23) | -.2436 (32.12) | -.3187 (41.44) | -.6001 (47.42) |
| /cut1 | -1.5064 | -1.0877 | -1.2179 | -1.0989 | 9.7357 |
| /cut2 | .1401 | .5004 | .3513 | .4309 |  |
| /cut3 | .9109 | 1.2902 | 1.1253 | 1.3203 |  |
| Pseudo/Adjusted $R^2$ | .0513 | .0414 | .0490 | .0416 | .1227 |
| N | 400,360 | 399,790 | 399,738 | 399,589 | 398,246 |
| Mean dependent variable | 2.05 | 1.78 | 1.90 | 1.81 | 7.15 |

Notes: In columns 1–4 ordered logits are run given the ordinal nature of the dependent variables. They are 4-step variables running from 'not at all' to 'nearly every day'. The 3 cut-points in the table are coefficients estimated by the model showing where the latent variable underlying the distribution of the dependent variable is cut to make 4 categories. The final column is an Ordinary Least Squares estimator which is appropriate for modelling the continuous additive scale. Controls are age, education, race, state, and week of interview.

The focus is the four-way COVID status variable at the top of the table. The base reference category is the half of individuals who have never had COVID. The three states in the table are having had short COVID only, having had long COVID but not currently suffering symptoms and finally those with long COVID and current symptoms.

In Table 5, because of the way the data is coded, a higher number implies worse mental health. In all five cases the pattern is the same: short COVID has a negative coefficient versus the excluded category of no COVID. Having long COVID is worse especially with current symptoms. The male coefficient is interesting in its own right, confirming that males are less likely to suffer from negative affect that females, confirming earlier work [23]. Working is negatively correlated with negative affect.

## 4.4: The impact of long COVID on wellbeing among the vaccinated and unvaccinated

In Section Two we reviewed evidence indicating vaccination reduced the likelihood of long COVID. But does it affect individuals' wellbeing? Bagues and Dimitrova [17] found that having been vaccinated raised well-being using UK data.

In our data we found little difference in vaccine rates across our four COVID states. 84.2% of those who have never had COVID had been vaccinated, although we don't know how many shots they have had. This compares with 82.0% of those with short COVID; 81.2% for those with long COVID but not presently with symptoms and 83.1% of those with long COVID now and 83.2% overall. As regards the link between vaccination and wellbeing we find similar results to Bagues and Dimitrova [17] although the extent of the wellbeing improvement varies across our four COVID type categories.

Below we show using the weeks #46-#53 data, that anxiety is higher if the respondent had not been vaccinated. The biggest differences are for those reporting anxiety nearly every day. Even among those who have had a vaccine and never had COVID 14% are anxious nearly every day versus 22% who had not been vaccinated. Results are similar using the other three affect variables.

**Table 6. Ordered logit estimates, physical mobility, remembering and understanding.**

|  | Mobility. | Dressing & bathing | Remember | Understand |
|---|---|---|---|---|
| Short COVID | -.3135 (33.00) | -.4509 (28.00) | -.1560 (20.76) | -.3970 (23.81) |
| Long COVID not now | .3115 (19.39) | .1003 (4.01) | .5660 (42.66) | .2302 (9.28) |
| Long COVID now | .9845 (70.39) | .7193 (35.61) | 1.3400 (102.45) | .9142 (45.57) |
| Male | -.2706 (33.23) | .0638 (4.84) | -.2657 (39.51) | +.3006 (22.42) |
| In paid work | -.7121 (78.13) | -1.1444 (76.55) | -.2871 (35.60) | -.5967 (38.79) |
| /cut1 | 1.1459 | .6125 | -.9711 | .3511 |
| /cut2 | 3.1538 | 2.6589 | 1.6389 | 2.7177 |
| /cut3 | 5.6540 | 4.5017 | 5.1847 | 4.4657 |
| Pseudo $R^2$ | .1210 | .0698 | .1420 | .0544 |
| N | 390,904 | 390,999 | 388,552 | 391,058 |
| Mean dependent variable | 1.34 | 1.11 | 1.52 | 1.10 |

Notes: Controls are age, education, race, state, and week of interview.

## 4.5: The impact of long COVID on aspects of daily life

Table 6 estimates equivalent equations to those in Table 5 but this time for four other potential outcomes of long COVID. The first two are difficulty walking or climbing stairs, what we term 'mobility' (column 1), and difficulties with self-care such as washing or dressing (column 2) while the other two relate to cognition—difficulties remembering or concentrating (column 3) and difficulties understanding or being understood (column 4). The pattern of results is identical to those presented in Table 5: compared to those who had never had COVID, short COVID is better than no COVID, whereas long COVID especially with current symptoms

| Anxious | No vaccine | Vaccinated |
|---|---|---|
| **Non COVID** | | |
| Not at all | 39 | 42 |
| Several days | 27 | 32 |
| More than half the days | 12 | 11 |
| Nearly every day | 22 | 14 |
| **Short COVID** | | |
| Not at all | 39 | 43 |
| Several days | 32 | 35 |
| More than half the days | 12 | 11 |
| Nearly every day | 17 | 12 |
| **Long COVID not now** | | |
| Not at all | 26 | 26 |
| Several days | 33 | 38 |
| More than half the days | 16 | 15 |
| Nearly every day | 24 | 22 |
| **Long COVID now** | | |
| Not at all | 17 | 17 |
| Several days | 30 | 35 |
| More than half the days | 16 | 17 |
| Nearly every day | 37 | 31 |

**Table 7. Ordered logit estimates, physical mobility, remembering and understanding and mental health.**

|  | Mobility. | Dressing & bathing | Remember | Understand |
|---|---|---|---|---|
| Short COVID | -.2163 (22.17) | -.3048 (18.01) | -.0163 (2.04) | -.2534 (14.76) |
| Long COVID not now | .2423 (14.68) | -.0007 (0.03) | .4894 (35.22) | .1348 (5.25) |
| Long COVID now | .7609 (52.92) | .4072 (19.23) | 1.0651 (79.15) | .6299 (30.13) |
| Male | -.2114 (25.31) | .1467 (10.72) | -.1747 (24.55) | +.4036 (29.11) |
| In paid work | -.6366 (68.18) | -1.0206 (66.12) | -.1675 (19.72) | -.4458 (28.17) |
| Combination mental health score | .1757 (146.72) | .2235 (128.91) | .2700 (239.22) | .2099 (119.09) |
| /cut1 | 3.0046 | 3.0017 | 1.4758 | 2.5060 |
| /cut2 | 5.1281 | 5.1375 | 4.4989 | 4.9476 |
| /cut3 | 7.6937 | 7.0153 | 8.1891 | 6.7211 |
| Pseudo $R^2$ | .1604 | .1442 | .1420 | .1221 |
| N | 388,586 | 388,597 | 388,552 | 388,729 |

Notes: Controls are age, education, race, state, and week of interview.

generates a significantly higher probability of facing the two physical and two mental health difficulties.

The possibility exists that some of the impact of long COVID on mobility and cognition could arise due to poor mental health. Indeed, there is evidence from Wang et al. [24] that prior psychological distress before SARS-CoV-2 infection is associated with risk of COVID–related symptoms lasting 4 weeks or longer. Hence Table 7 uses the same four outcomes as in Table 6 but now also includes the mental health score as a control variable which is the sum of the negative affect variables. Unsurprisingly, scoring high on mental health problems is associated with problems with mobility, dressing and bathing, remembering and understanding. However, even though its incorporation results in a decline in the size of the coefficients attached to COVID the COVID variables are of the same sign as in Table 6 and remain highly statistically significant.

## 5. Conclusions

We have exploited new data for nearly half a million Americans on the prevalence of long COVID to explore its incidence and correlates, and the relationship between long COVID and physical and mental health problems. Long COVID is widespread. It has affected 14 percent of Americans at some point.

Its incidence varies markedly across sub-groups in the population. It is much higher among women than it is among men; it varies by ethnicity, being highest among whites; and it is hump-shaped in age, mimicking the age profile of negative affect which is highest in middle age. It also varies greatly by location. It is highest in states in the south such as West Virginia and Mississippi–states where negative affect is high–and is lowest in Hawaii, notable for being the 'happiest' state in the United States.

Long COVID is independently associated with negative affect, however one measures it, and with physical mobility and mental health problems. These associations are strongest among those who report current symptoms of long COVID.

Notwithstanding these new findings much remains to be learned about the nature, determinants and consequences of long COVID which will only be revealed in time with the advent of new data. In particular, exploiting longitudinal data tracking individuals over time could be particularly informative since in the current study the cross-sectional nature of the data makes it hard to make causal inferences about the impact of Long COVID and the potential value of vaccinations.

## Supporting information

**S1 Appendix. Questions used to identify long COVID–with weighted percentages in square parentheses.**
(DOCX)

## Author Contributions

**Conceptualization:** Alex Bryson.

**Data curation:** David G. Blanchflower.

**Formal analysis:** David G. Blanchflower, Alex Bryson.

**Methodology:** David G. Blanchflower, Alex Bryson.

**Writing – original draft:** David G. Blanchflower, Alex Bryson.

**Writing – review & editing:** David G. Blanchflower, Alex Bryson.

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
