## [Decision Letter · Decision Letter 0]

4 May 2023

PONE-D-23-04744Long COVID in the United StatesPLOS ONE

Dear Dr. Bryson,

Thank you for submitting your manuscript to PLOS ONE. After careful consideration, we feel that it has merit but does not fully meet PLOS ONE’s publication criteria as it currently stands. Therefore, we invite you to submit a revised version of the manuscript that addresses the points raised during the review process. Please organize the paper in standard PLOS one format and review additional papers on the same field. Use appropriate referencing throughout the paper. 

We look forward to receiving your revised manuscript.

Kind regards,

Kanchan Thapa, MPH, MPhil

Academic Editor

PLOS ONE

2. In your Methods section, please include additional information about your dataset and ensure that you have included a statement specifying whether the collection and analysis method complied with the terms and conditions for the source of the data.

“NO - Include this sentence at the end of your statement: The funders had no role in study design, data collection and analysis, decision to publish, or preparation of the manuscript”

Additional Editor Comments:

Dear authors,

I enjoyed reading your paper. I suggest you to revise your paper adhering to reviewer’s comment and also with the following suggestions:

1. Please include the referencing of each item that you have cited in the paper. Without referencing, we can’t move ahead

2. Please organize your paper in the simple, scientific, fluent, and understandable language

3. Please follow the format of a standard research paper while organizing your paper

4. It seems like you have uploaded the draft version of the paper, so I suggest you to upload the final version of the paper clearly stating: background, methods, results, discussion/ conclusion section. I suggest you to review additional paper on PLOS One for details.

5. Try to improve your paper based on reviewer I.

Reviewers' comments:

Reviewer's Responses to Questions

**Comments to the Author**

1. Is the manuscript technically sound, and do the data support the conclusions?

Reviewer #1: Yes

Reviewer #2: Partly

2. Has the statistical analysis been performed appropriately and rigorously? 

Reviewer #1: Yes

Reviewer #2: I Don't Know

3. Have the authors made all data underlying the findings in their manuscript fully available?

Reviewer #1: Yes

Reviewer #2: No

4. Is the manuscript presented in an intelligible fashion and written in standard English?

Reviewer #1: Yes

Reviewer #2: No

5. Review Comments to the Author

Reviewer #1: the article is well written

age description and distribution among tables are not unified

it would have been interesting to showcase symptoms vs annual income rate

a comparison between vaccinated vs non vaccinated group would be interesting

Reviewer #2: This is an interesting paper looking at an important topic: long COVID. The paper computes prevalence overall and by demographic and geographical levels. It also describes how long covid is associated with negative affect for different demographic groups and by vaccination status.

In my view, the paper is too difficult to read and it is therefore complicated to understand how sound the description is. The paper generally lacks context and depth in discussing the findings by subgroups and is not suited for an international multidisciplinary audience.

The definition of short and long covid is not given.

Data are described with sweeps only instead of giving the corresponding period covered.

Literature review unclear:

-Regarding Ayoubkhani's findings (page 2), it is unclear whether the second dose was less effective than the first dose. This needs be clarified or rephrased if necessary.

- On page 4, the authors claim that their results are consistent, but they only cite one study without providing context. Was this study also conducted in the US ?

The presentation of tables and charts are challenging:

Table 3 raises the question of whether certain groups were excluded from the analysis. It is unclear if these excluded groups are the reference categories or if they were excluded for other reasons. For example, what is happening with the 89 years old and over category if those under 20 are the reference category? The layout of the tables is poor and the discussion of the construction of the results presented in the tables is so absent that it is difficult to assess the reliability of the interpretation made in the main text.

What is the sample size in Table 4?

Chart 1 plots numbers. There is no context neither on the chart nor in the text.

No URL for data availability. Manuscript submission requirements were not met, eg. double spacing, footnotes etc. No line numbers to refer to for comments.

6. PLOS authors have the option to publish the peer review history of their article (what does this mean?). If published, this will include your full peer review and any attached files.

Reviewer #1: No

Reviewer #2: No

---

## [Author Response · Author response to Decision Letter 0]

13 Jun 2023

PONE-D-23-04744

Long COVID in the United States

PLOS ONE

13th June 2023

Dear Dr. Kanchan Thapa,

Thank you for giving us the opportunity to revise our manuscript for PLOS ONE. 

This is our letter responding to each point raised by the academic editor and reviewers. We also attach a marked-up copy of our manuscript that highlights changes made to the original version together with an unmarked version of our revised paper without tracked changes. 

We have tried to meet PLOS ONE's style requirements, including those for file naming. 

In our section entitled “Data and Methods” we have included additional information about the dataset and how to access it.

You have asked us to clarify the sources of funding (financial or material support) for your study. There were none. So we can confirm that we received no funding for this project.

The referencing now follows the standard approach in PLOS ONE with numerical indicators in squared brackets pointing to the relevant reference in the bibliography.

In terms of style and content we have organised the paper in a simple fashion adopting the structure that is standard in PLOS ONE. In doing so we have modified our language to make it more scientific, fluent and direct.

The sections are now formatted in a way that is standard for PLOS ONE papers.

Below we outline how we have responded to the reviewers’ comments. Our responses are in underlined bold italics.

Reviewer #1

1. The article is well written

RESPONSE: Thank you.

2. age description and distribution among tables are not unified

RESPONSE: We wish to retain the differences in the age classifications in Tables 2 and 3. Table 2 is more aggregated because we are providing a descriptive overview but we want to be more granular in Table 3 in the regression analysis. 

3. it would have been interesting to showcase symptoms vs annual income rate

RESPONSE: Although this additional analysis could be interesting the paper is quite long already so we don’t think there is much room for additional heterogeneity analysis. Instead we have focused on key differences, namely when the respondent had long COVID and the extent of the symptoms and the role of the vaccine.

4. a comparison between vaccinated vs non vaccinated group would be interesting

RESPONSE: Again, this is something one could have added, but it isn’t central to our concerns which are about the correlates of long COVID – including vaccine receipt - and the association between long COVID and wellbeing outcomes. Given the length of the paper we have chosen to omit analysis of differences between those who have been vaccinated and those who have not. 

Reviewer #2

This is an interesting paper looking at an important topic: long COVID. The paper computes prevalence overall and by demographic and geographical levels. It also describes how long covid is associated with negative affect for different demographic groups and by vaccination status.

In my view, the paper is too difficult to read and it is therefore complicated to understand how sound the description is. The paper generally lacks context and depth in discussing the findings by subgroups and is not suited for an international multidisciplinary audience.

RESPONSE: We have tried to simplify the structure and language.

The definition of short and long covid is not given.

RESPONSE: We provide a definition of long COVID in paragraph 2 but return to the issue in the literature review pointing out that definitions vary a little across studies. Appendix 1 details the questions used in our data to define long COVID. We define short COVID as having COVID symptoms for less than 3 months.

Data are described with sweeps only instead of giving the corresponding period covered.

RESPONSE: We provide dates for each sweep in footnote 1 and provide dates in various places.

Literature review unclear:

-Regarding Ayoubkhani's findings (page 2), it is unclear whether the second dose was less effective than the first dose. This needs be clarified or rephrased if necessary.

RESPONSE: The authors do not speculate as to whether a second dose was less effective than the first. Instead, they couch their findings in terms of the second dose sustaining the protection against long COVID. We have added a phrase in to that effect.

- On page 4, the authors claim that their results are consistent, but they only cite one study without providing context. Was this study also conducted in the US?

RESPONSE: The other paper is also for the USA. We now make this clear in the text.

The presentation of tables and charts are challenging:

Table 3 raises the question of whether certain groups were excluded from the analysis. It is unclear if these excluded groups are the reference categories or if they were excluded for other reasons. For example, what is happening with the 89 years old and over category if those under 20 are the reference category? The layout of the tables is poor and the discussion of the construction of the results presented in the tables is so absent that it is difficult to assess the reliability of the interpretation made in the main text.

RESPONSE: The ‘excluded’ categories are indeed the reference groups for the regression. The footnote has been revised to make this clear. There is nobody over the age of 89.

What is the sample size in Table 4?

RESPONSE: Sample size is 454,706 which has been added to the tabl

Chart 1 plots numbers. There is no context neither on the chart nor in the text.

RESPONSE: The Chart (relabelled Fig. 1) provides context by showing the incidence of COVID cases and deaths reported each week for the U.S. by the CDC. The figure is described in the first paragraph. We have relabelled the figure so it is clear what is being presented.

No URL for data availability. 

RESPONSE: the URL for the data is now provided. It is publicly available.

Manuscript submission requirements were not met, eg. double spacing, footnotes etc. No line numbers to refer to for comments.

RESPONSE: The main text is now double spaced and all footnotes have been removed and line numbers added.

---

## [Decision Letter · Decision Letter 1]

25 Jul 2023

PONE-D-23-04744R1Long COVID in the United StatesPLOS ONE

Dear Dr. Bryson,

Thank you for submitting your manuscript to PLOS ONE. After careful consideration, we feel that it has merit but does not fully meet PLOS ONE’s publication criteria as it currently stands. Therefore, we invite you to submit a revised version of the manuscript that addresses the points raised during the review process.

I am echoing with reviewer comments. Please revise the paper based on their comments. Also, review, revise and resubmit. Please also make changes based on previous rounds of comments by reviewers. 

We look forward to receiving your revised manuscript.

Kind regards,

Kanchan Thapa, MPH, MPhil

Academic Editor

PLOS ONE

Additional Editor Comments:

Dear Authors,

I enjoyed reading your revised paper. Reviewing the paper and echoing with reviewers, I would like to request you to revise the tables and take care of the comments raised in before too. Authors should address each comments properly especially those raised on Table and interpretation section. Following the second round of review too, a reviewer made similar comments on interpretation and table. Please revise and resubmit the paper.

Be consistent with the authors who contributed the paper? List all the authors in the system.

Similarly, authors mentioned COVID throughout the paper. Is there any difference between COVID 19 and COVID? IS there any competing interest to mention COVID-19 by the authors? Please also ensure both authors read the paper, revise based on your review and submit the paper.

Reviewers' comments:

Reviewer's Responses to Questions

**Comments to the Author**

1. If the authors have adequately addressed your comments raised in a previous round of review and you feel that this manuscript is now acceptable for publication, you may indicate that here to bypass the “Comments to the Author” section, enter your conflict of interest statement in the “Confidential to Editor” section, and submit your "Accept" recommendation.

Reviewer #3: (No Response)

Reviewer #4: (No Response)

2. Is the manuscript technically sound, and do the data support the conclusions?

Reviewer #3: Yes

Reviewer #4: Partly

3. Has the statistical analysis been performed appropriately and rigorously? 

Reviewer #3: Yes

Reviewer #4: I Don't Know

4. Have the authors made all data underlying the findings in their manuscript fully available?

Reviewer #3: Yes

Reviewer #4: Yes

5. Is the manuscript presented in an intelligible fashion and written in standard English?

Reviewer #3: Yes

Reviewer #4: No

6. Review Comments to the Author

Reviewer #3: This revised manuscript is well written and addresses prevalence of long COVID in different level, i.e., demographic and geographical levels, as well as the relationship between long COVID and physical and mental health problems.

Compared to the original manuscript, it is more organized and well presented.

Reviewer #4: The findings as presented in the tables are difficult to interpret at best. It appears to omit key, usual statistical output. The presentation of the tables are lacking, e.g. identification of what is presented and of measures of statistical significance.

What are the authors' views about their findings? - e.g., on short COVID associated with least MH and well-being risk? Why are those who are middle age and female most affected?

7. PLOS authors have the option to publish the peer review history of their article (what does this mean?). If published, this will include your full peer review and any attached files.

Reviewer #3: No

Reviewer #4: No

---

## [Author Response · Author response to Decision Letter 1]

28 Jul 2023

PDF Page, Line Comment Response

43 - Abstract It “peaks in midlife” -Can we refer to long covid as peaking given the relatively short duration of this event, not allowing for an appropriate follow-up period? “peaks in midlife” has been replaced with “is at its highest in midlife”. 

45, 28 The paragraph in the Introduction is a repeat of the Abstract. What’s the purpose? Here again is the phrase about peaking in midlife. The purpose is simply to summarise the paper and its results in the main body of the text early on. “peaks in midlife” has been replaced with “is at its highest in midlife”.

45, 33-34 “The effect is larger …” What is being referred to here, is it long COVID (LC)? And which groups are being compared? Is it the effect of LC is larger among people who currently report LC compared to those who experienced LC in the past? “The effect is larger” has been replaced by “with the correlation being strongest among those who currently report long COVID, especially if they report severe symptoms”.

46, 48-50 Italics and quotation marks are not used in a consistent way. Quotations are no longer italicised

46, 55 The authors’ recap of O’Mahoney et al’s findings – is ‘COVID-19 survivors’ the same as anyone who had COVID? Yes that’s what it means. We have left it unchanged because it seems clear from the text.

46, 62-63 Results of Taquet are confusing – was a larger proportion of survivors (57%) affected after 180 days than in the earlier period (37% in 90 to 180 days)? The sentence has been rephrased to read “57% of patients having at least one long-COVID feature recorded within the first 180 days after infection and 37% having them in the 90 to 180 days after diagnosis”

46, 65 > (paragraph) Is there a reason for this paragraph to be written in the present tense? The paragraph has been revised so that it uses the past tense.

47, 79 > 

47, 84 Rather than per population, could we report on the per cent of the survivors with long COVID?

‘They adversely impacted …’ is ‘they’ referring to long COVID? The earlier use of ‘they’ in that paragraph referred to the Office of National Statistics. They express the incidence relative to the population so this is what we report.

We have revised the text to read “Long COVID symptoms adversely impacted the day-to-day activities of 1.6 million”. Thank you for spotting this.

52, 185 till Wed 

59, 345 Is this a table of results from the current study? The content focuses on anxiety. However, I do not recall details about the measure of anxiety and how the scores are interpreted in the Methods section. 58, 318 – seems to be referring to mental health data and coding, which should be expanded in the Methods. Yes the table is from our own analysis. The details of the anxiety measure are provided a little earlier in the article and are replicated here for reference. We use a) for anxiety from the following question:

Over the last 2 weeks, how often have you been bothered by:

a) feeling nervous, anxious, or on edge? 

b) not being able to stop or control worrying?

c) by feeling down, depressed, or hopeless?

d) bothered by having little interest or pleasure in doing things?

Answers were coded as Not at all =1, Several days =2, More than half the days=3 and Nearly every day=4.

Rather than having a very lengthy methods section incorporating all questions we have decided to report them as we present our results, so that the reader finds it easier to refer to them when absorbing the results.

59, 342 Is the 15% referring to the 14% in the data presented on anxiety? … in the ‘non COVID’ group. Thank you for spotting this. The text has been revised to read: “Even among those who have had a vaccine and never had COVID 14% are anxious nearly every day versus 22% who had not been vaccinated”

60, 374 “The four long COVID categories each have positive signs..” unclear what this means. Thank you for this. We have clarified the meaning of the sentence by rephrasing it as follows: “Having long COVID now, or in the past, is associated with higher negative affect compared to having no COVID and no vaccine but among those with long COVID now or in the past, negative affect was always lower if they had had the vaccine.”

 Table 5 Notes – it is unclear what is meant by ‘age dummies’ and in subsequent tables ‘Working/cut1/ cut2 etc. These are not explained or described. The word “dummies” has been dropped from all table footnotes so that we now refer to controlling for age. “Working” has been replaced by “In paid work” to denote the respondent’s labour market status. Cut1/cut2/cut3 are part of the standard output from an ordered probit regression so these are retained

 Table 6 – conventional formatting of this and other tables might help the reader to understand what is being presented. The models are presented in a conventional fashion. We assume that the comments and changes above clarify what is presented.

 Chart 1 differ from Figure 1? We only refer to Figure 1 now. There is no Chart 1

61, 389 Isn’t the reference to Wang et al discussing what this paper defines as ‘long COVID’? What is ‘post-COVID-19 conditions” and is the ‘?’ a typo? We have clarified what the sentence means so it reads: “There is evidence from Wang et al. [24] that prior psychological distress before SARS-CoV-2 infection is associated with risk of COVID–related symptoms lasting 4 weeks or longer.”

Thanks yes the question mark was a typographical error.

61-62 The authors’ interpretation of the findings is not provided; there is no Discussion. The Conclusions section provides a summary of the findings. Yes, we simply summarise the findings from our study in the conclusion because these are new findings about the prevalence, correlates and wellbeing-related outcomes linked to long-COVID which are not well-documented. However, we have added the following sentence: “Notwithstanding these new findings much remains to be learned about the nature, determinants and consequences of long COVID which will only be revealed in time with the advent of new data.”

---

## [Decision Letter · Decision Letter 2]

21 Sep 2023

PONE-D-23-04744R2Long COVID in the United StatesPLOS ONE

Dear Dr. Bryson,

Thank you for submitting your manuscript to PLOS ONE. After careful consideration, we feel that it has merit but does not fully meet PLOS ONE’s publication criteria as it currently stands. Therefore, we invite you to submit a revised version of the manuscript that addresses the points raised during the review process. Please address the comments from the reviewers. 

We look forward to receiving your revised manuscript.

Kind regards,

Kanchan Thapa, MPH, MPhil

Academic Editor

PLOS ONE

Journal Requirements:

Additional Editor Comments:

Dear Authors,

Please address the few more changes requested by peer reviewers.

Reviewers' comments:

Reviewer's Responses to Questions

**Comments to the Author**

1. If the authors have adequately addressed your comments raised in a previous round of review and you feel that this manuscript is now acceptable for publication, you may indicate that here to bypass the “Comments to the Author” section, enter your conflict of interest statement in the “Confidential to Editor” section, and submit your "Accept" recommendation.

Reviewer #2: All comments have been addressed

Reviewer #5: All comments have been addressed

2. Is the manuscript technically sound, and do the data support the conclusions?

Reviewer #2: Yes

Reviewer #5: Yes

3. Has the statistical analysis been performed appropriately and rigorously? 

Reviewer #2: Yes

Reviewer #5: Yes

4. Have the authors made all data underlying the findings in their manuscript fully available?

Reviewer #2: Yes

Reviewer #5: (No Response)

5. Is the manuscript presented in an intelligible fashion and written in standard English?

Reviewer #2: Yes

Reviewer #5: Yes

6. Review Comments to the Author

Reviewer #2: please amend line 371: what is "comp"?

Please mention in titles when tables are only descriptive (like Table 4). The way it currently stands, we are not sure what Table 4 is, they should be self explanatory.

Could people with long covid had their vaccination after getting long covid? If so, I dont know what to do with all the vaccination part. If people get vaccinated (or not) because of getting long covid, splitting the sample by vaccination status does not seem to bring additional information.

I agree with the other reviewer that the standard output from ordered logit such as cut1 cut2 etc need to be explained at least in the table notes .

People who dont get COVID may be comprised of those who got out, some of which got covid, and people who didnt get out and didnt get covid. This would explain why those who didnt get covid seem worse off. I wonder if potential explanations for the results could go in the conclusion or discussion. The paper generally lacks a section on limitation.

Reviewer #5: Summarize the second paragraph of introduction part and merge it with first paragraph of introduction part (Long Covid Definition).

7. PLOS authors have the option to publish the peer review history of their article (what does this mean?). If published, this will include your full peer review and any attached files.

Reviewer #2: No

Reviewer #5: No

---

## [Author Response · Author response to Decision Letter 2]

22 Sep 2023

Dear Editor

Response to Reviewers PONE-D-23-04744R2 “Long COVID in the United States”

We thank you for giving us a further opportunity to revise the paper. Below we explain in bold italics how we have responded to the points made by the reviewers.

Reviewer #2: 

please amend line 371: what is "comp"?

Thank you for spotting this error. We have revised the text accordingly.

Please mention in titles when tables are only descriptive (like Table 4). The way it currently stands, we are not sure what Table 4 is, they should be self explanatory.

We have amended the table title accordingly.

Could people with long covid had their vaccination after getting long covid? If so, I dont know what to do with all the vaccination part. If people get vaccinated (or not) because of getting long covid, splitting the sample by vaccination status does not seem to bring additional information.

We have removed table 6 accordingly, together with the attendant text and comments in the conclusion.

I agree with the other reviewer that the standard output from ordered logit such as cut1 cut2 etc need to be explained at least in the table notes .

We have extended the table notes accordingly.

People who dont get COVID may be comprised of those who got out, some of which got covid, and people who didnt get out and didnt get covid. This would explain why those who didnt get covid seem worse off. I wonder if potential explanations for the results could go in the conclusion or discussion. The paper generally lacks a section on limitation.

We have noted the limitations to our paper due to the cross-sectional nature of the data and indicated why advances in knowledge could come from longitudinal data tracking individuals over time. In the absence of such data we have decided not to speculate about the relative mental health of those who don’t get COVID versus those who get short COVID. 

Reviewer #5

Summarize the second paragraph of introduction part and merge it with first paragraph of introduction part (Long Covid Definition).

We have merged these paragraphs together.

---

## [Editor Report · Decision Letter 3]

27 Sep 2023

Long COVID in the United States

PONE-D-23-04744R3

Dear Dr. Bryson,

We’re pleased to inform you that your manuscript has been judged scientifically suitable for publication and will be formally accepted for publication once it meets all outstanding technical requirements.

Kind regards,

Kanchan Thapa, MPH, MPhil

Academic Editor

PLOS ONE
---

## [Editor Report · Acceptance letter]

6 Oct 2023

PONE-D-23-04744R3 

Long COVID in the United States 

Dear Dr. Bryson:

I'm pleased to inform you that your manuscript has been deemed suitable for publication in PLOS ONE. Congratulations! Your manuscript is now with our production department. 

Kind regards, 

on behalf of

Mr. Kanchan Thapa 

Academic Editor

PLOS ONE